# Small-Scale Marine Fishers' Possession of Fishing Vessels and Their Impact on Net Income Levels: A Case Study in Takalar District, South Sulawesi Province, Indonesia

**Ahmad Imam Muslim** [1,*], **Miho Fujimura** [2], **Tsuji Kazunari** [2] **and Muslim Salam** [3]

1   The United Graduate School of Agricultural Sciences, Kagoshima University, Kagoshima 890-8580, Japan
2   Faculty of Agriculture, Saga University, Saga 840-8502, Japan; fujimum@cc.saga-u.ac.jp (M.F.); tsujikjp@cc.saga-u.ac.jp (T.K.)
3   Laboratory of Farm Management and Agricultural Marketing, Department of Socio-Economics of Agriculture, Faculty of Agriculture, Hasanuddin University, Makassar 90245, Indonesia; muslimsal@yahoo.com
*   Correspondence: aim_imam.kun46@yahoo.com

**Abstract:** Over the last two decades, the growth of the fisheries sector in Indonesia has shown an increasing trend; however, behind the rapid development of this sector, the role of small-scale fisheries as one of the main actors supporting the whole industry is often neglected. They remain poor and continue to conduct fishing activities traditionally. Therefore, this study aims to describe the real situation of small-scale fishers, analyzing the fishers' ownership of the boat they use, analyzing their income level to reveal their poverty status, and analyzing the factors affecting their income. The regression analysis results indicate that boat category, sea fish catch, fish selling price, fixed costs, and variable costs have significant effects on fishers' net incomes.

**Keywords:** small-scale fisher; fishers' boat; net income level of fishers; rowboat; outboard motorboat; motorboat

**Key Contribution:** Small-scale fishers' contribution to the fisheries sector at large cannot be neglected, but they are in poor condition. One of the reasons for this is their traditional fishing vessels, which affect income. In order to make a better decision on how to develop their livelihoods, all of the internal and external stakeholders (fishers, the fisheries department, government, NPO, NGO, and funding agencies) should work together to establish a system of cooperation between fisheries, a fisheries association, and to create a chance for fishers to diversify their livelihoods.

## 1. Introduction

The development of the fisheries sector in Indonesia plays an important role in terms of employment creation, food security, poverty alleviation, and economic development [1–3]. The development of the total fisheries production has shown an increasing trend, especially over the last two decades, which has made Indonesia the second major producer of fisheries and aquaculture products in the world [4]. Despite this rapid development, it still has many challenges to overcome, for instance, improving the livelihood of small-scale fishers. The role of small-scale fishers is important because they are the suppliers of seafood to people in coastal and inland societies and also absorb labor in order to alleviate poverty problems in remote areas [5–7].

There were approximately 5.9 million fishers and fish farmers in Indonesia in 2016, and there were around 960,000 households engaging in capture fisheries [8,9]. Although these households are the ones who made Indonesia the world's second-largest producer of fishery products, the fact remains that, among Indonesia's poor people, one-fifth originate from fishery households [9]. Most likely, these households are supported by small-scale fishers. Even though small-scale marine capture fishers in Indonesia are the largest contributors to

domestic fishery production, about 85% of the workforce engaged in the fishing sector is still made up of poor and traditional fishers. They are lagging behind in terms of education, fishing skills, and fishing activity management, with limited access to other livelihood choices, a lack of capital, and high dependency on natural resources, resulting in their poor economic condition [10–13]. As stated in the Indonesian Constitution No. 45/2009 Article 1 point 11 [14], small-scale fishers are defined as people who catch fish as part of their livelihood and use a boat with a maximum of 5 GT in order to fulfill their daily needs. People who are catching fish and using a boat that does not exceed 5 GT is a clear statement and accessible; however, the remaining question is whether these small-scale fishers can actually fulfill their daily needs or whether they are instead living a difficult life. Thus, addressing the poverty of small-scale fishers remains an important issue for the Indonesian fisheries sector.

Despite the importance of small-scale fishers, they experience poor economic conditions because of a low-income level. The income level of fishers can be affected by many factors. This is according to Agunggunanto and Arianti [15], who used ordinary least square (OLS) analysis, education level, boat ownership, fishers' assets, cooperation aid, and fish selling as significant factors affecting the income level of fishers in the Demak District, Central Java. The income level of fishers who own a boat is higher because they do not need to make a contract with a boat owner like fishers who do not have a boat. Furthermore, Putra [16] mentioned that the bigger the size of the boat engine, the higher the fishers' income. The type of boat is an important aspect of fishing activities for fishers because owning an outboard motorboat or motorboat enables fishers to travel to further fishing grounds, which can lead to a higher fish catch and ultimately increase their income. As shown by the research of Suri and Kune [17], which was conducted in the East Nusa Tenggara Province, the net income of fishers with a motorboat is almost three times the monthly net income of a fisher with an outboard motorboat. In addition, Rahim and Hastuti [18] observed that the income of fishers with a row boat was Rp. 191,000/trip, and that of fishers with an outboard motorboat was Rp. 468,066/trip among fishers in the Barru District, South Sulawesi Province. In accordance, the power of the engine of the boat positively affects the income of fishers in the Takalar District and Barru District, South Sulawesi Province, which means that the higher the engine power of the boat, the higher the income of fishers in this area [19,20].

Based on the elucidation above, it can be inferred that a determinant impacting the financial earnings of fishers is the type of vessel utilized for fishing purposes. The fishing vessel is a crucial instrument for fishermen's fishing operations. The legal definition also refers to the fishing boat; therefore, possession or ownership of a boat is very important. There has been a noticeable upward trajectory in the ownership of fishing boats among those engaged in the fishing industry in Indonesia in recent years. The Indonesian government has established many programs in order to improve fishers' livelihoods and well-being, and one of these programs was based on Presidential Decree No. 39 1980 about a small-scale fishers' boat motorization program in order to increase fishers' productivity [21]. Subsequently, this program has persisted. The influence of the governmental program is shown in Figure 1. The use of row boats among fishers has experienced a fall, while the adoption of outboard motorboats and motorboats has significantly increased since 2003 (Figure 1) [22].

Furthermore, it is worth noting that there has been a notable increase in the national capture production volume, which could be attributed to advancements in boat technology. Specifically, the production volume has risen from 4.9 million tonnes in 2000 to 7.7 million tonnes in 2019 [23]. A corresponding increase in fishers' incomes accompanied the enhancement of fishing vessel types. Based on the findings of MMAFI [24], it was observed that in 2014, the mean monthly income of those engaged in fishing activities amounted to Rp. 2,150,000 (equivalent to USD 156, with an exchange rate of USD 1 = RP 13,771). Subsequently, in 2017, this figure had an upward trend, reaching RP 2,700,000 (equivalent to USD 196) per month. A potential correlation exists between the increase in fishing

vessel ownership and the subsequent rise in fishermen's catch productivity, resulting in an increased income.

Based on the preceding discourse, it is evident that the fisheries sector holds significant importance for Indonesia, as seen by recent advancements in the acquisition of fishing vessels by fishers and the corresponding increase in their incomes. Regrettably, there exists a dearth of research examining the actual circumstances faced by small-scale fishers concerning their fishing endeavors, the ownership of fishing vessels, socioeconomic impoverishment, and the determinants of income. Consequently, this study aims to elucidate the genuine conditions experienced by small-scale fishers, scrutinizing their possession of fishing boats, evaluating their income levels to ascertain poverty status, and investigating the factors that influence their earnings.

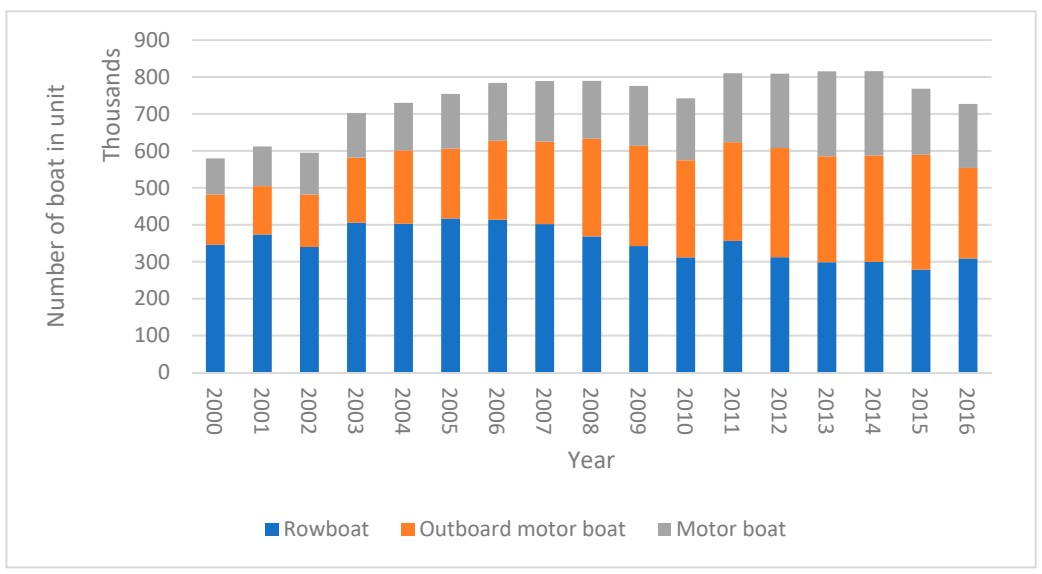

**Figure 1.** Boat possession of fishers in Indonesia [22].

## 2. Materials and Methods

The research site was in the Takalar District, one of the South Sulawesi Province districts. The Takalar District was chosen as the study region due to its significant proportion of coastal territories and a substantial population of fishermen. According to the information depicted in Figure 2, the geographical region marked by a red dot on the map corresponds to the Takalar district. Research was conducted in five selected sub-districts out of the nine sub-districts in Takalar District. These sub-districts included Mangarabombang, Sanrobone, Galesong, South Galesong, and North Galesong. The total area covered by these sub-districts was 240.88 km2, with a coastline spanning 74 km (Statistics Agency of Takalar District, 2017).

The present study employed a cross-sectional survey design. One hundred sixty-nine fishers were surveyed in the Takalar District, but only 152 fishers had completed data and were included in the subsequent data analysis. The approach employed for the selection of sample respondents was convenience sampling, which is often referred to as haphazard sampling or accidental sampling. Data gathering for this research encompassed two main types: primary data and secondary data collecting. Primary data collection involved conducting household interviews with fishers, which took place between February and March 2018. In order to obtain a complete and unbiased understanding of this issue, an unstructured interview was conducted with a key informant to gather firsthand information on fishers. This approach aimed to minimize biases and inaccuracies in the data collection process. The individuals who provided crucial information for this study included government officials, fishing association leaders, and social community leaders at the designated site. In total, there were six key informants. The interviews were of significant importance,

as they had the dual purpose of obtaining essential data and seeking authorization to access the study location, ensuring the seamless execution of the data collection process. Another factor contributing to this phenomenon is the inherent skepticism within such communities toward individuals seeking information about their domestic affairs. In order to facilitate the author's introduction to the fishers, it was suggested that community leaders or government officials take the initiative. The questionnaire underwent multiple revisions based on the findings obtained from key informant interviews. During the revision process, the author adjusted all the questions and created a comprehensive list encompassing fishing gear type, fishing net types, the potential costs associated with fishing activities, and other relevant aspects of their fishing endeavors. Prior to commencing the primary data collection, a pilot study was conducted to test the questionnaire with a sample of five individuals engaged in fishing activities. The individuals engaged in fishing activities were interviewed in accordance with the administered questionnaire. However, within the fishing equipment portion, the fishermen encountered difficulties when comprehending the terminology used by the author. Consequently, the author provided a visual aid in the form of a photo depicting the equipment and an explanation of the corresponding local term. In order to align with the vernacular often employed by local fishers in this region, certain sections of the questionnaire were modified. Subsequently, the questionnaire was revised to ensure its alignment with the specific fishing terminology used in the local context. Following this, the questionnaire was ultimately finalized for data collection.

The respondents of this study were purposively recruited from five specific sub-districts within the Takalar District, where they dwelt and were actively involved in maritime fishing activities. Marine fishers are individuals who engage in the capture of fish inside marine environments, specifically in seawater. The procedures employed to select the sample fishermen were as follows: Two villages located along the shoreline were randomly picked from each of the five sub-districts using a lottery method. Within each sub-district, the sample consisted of approximately 31-36 individuals engaged in fishing activities. This criterion served as a benchmark for determining the minimum sample size in each sub-district. Consequently, it was ensured that a sample size of 15-18 fishers was questioned in each hamlet.

Primary data were collected through the utilization of a standardized questionnaire survey. This questionnaire was structured into distinct sections: household background, fishing activity, household expenditure, and livelihood outcomes. The household background section encompassed variables such as the age of the fishers, their formal education background, and the number of household members. The fishing activity section included variables such as fishing experience, fishing days, boat possession, the number of fish caught, the fish selling price, fishing ground, number of net types possessed, and fishing cost. The household expenditure section focused on the financial aspects of the household, while the livelihood outcomes section examined both fishing and secondary income.

In statistical analysis, a set of 13 independent variables is commonly utilized. These variables include Age ($X_1$), School period ($X_2$), Fishing experience ($X_4$), Household members ($X_4$), Fishing days ($X_5$), Boat category ($X_6$), Sea fish catch ($X_7$), Fish selling price ($X_8$), Fishing ground ($X_9$), Total net ($X_{10}$), Fix cost ($X_{11}$), Variable cost ($X_{12}$), and Second job ($X_{13}$). The variable of interest in this study was net income, denoted as $Y$, which served as the dependent variable. The revenue of fishers was derived from the product, the average quantity of fish caught by fishers, the average number of fishing days in a month, and the selling price of fish. Based on the interviews conducted with individuals engaged in fishing activities, it was found that they followed a specific fishing routine. This routine entailed either engaging in small fishing trips daily or embarking on fishing expeditions lasting approximately 12–24 h each day for a consecutive period of four days, followed by a subsequent rest period of two days. The determination of the number of fishing days by fishers can be inferred from their established routines. The quantity of fish caught by Fishers was derived from the average daily catch of fishers. Most fishermen typically sell their catch to village collectors or money lenders, with the selling price remaining relatively stable

over three to six months and seldom experiencing fluctuations. Therefore, the accuracy of the selling price of fish might be deemed acceptable. Net income refers to the aggregate revenue derived from fishing operations and secondary employment, subtracting variable and fixed costs. The fishing variable costs in this context encompassed various expenses such as fuel, ice, labor, food, bait, daily net repair, and other incidental costs that arise during fishing operations. This research encompassed various fishing-related expenses, such as boat rental fees, engine rental fees, and maintenance expenditures. The acquired primary data were analyzed using descriptive statistics in MS EXCEL and SPSS 22 edition.

Secondary data collection involves sourcing information from previously published research and secondary data sources. These data were exclusively utilized to provide contextual background and elucidate the circumstances of small-scale fishers. The utilization of secondary data is not employed inside the analysis section. In the year 2015, the Takalar District was reported to have a total of 2085 individuals engaged in marine capture fishing activities [25]. In the Takalar District, the year 2016 witnessed the presence of 527 row boats, 2402 outboard motorboats, and 907 motorboats. Additionally, the catch fisheries in the district yielded a total production of 9372 tons [26]. Being situated along the coastline, the inhabitants of this district have historically engaged in maritime occupations such as seafaring and fishing. Before the 1970s, the fisheries industry in South Sulawesi Province operated predominantly traditionally, devoid of motorized fishing vessels and mechanized gear. In the specific context of Takalar District, historical records indicate that fishermen's adoption of engine-powered boats did not occur until approximately 1975. Subsequently, during the 1980s, a progressive modernization process was observed in fishing vessels and tools [27]. In the Takalar District, the poverty line was set at RP 299,721 per capita per month, equivalent to USD 0.73 per day (based on the exchange rate of USD 1 = RP 13,771). In 2017, the number of individuals living below the poverty line in the Takalar District was recorded as 26,990, accounting for 9.24% of the total population [28]. In order to evaluate the extent of poverty among fishers in this study, the poverty line criteria established by the World Bank were employed.

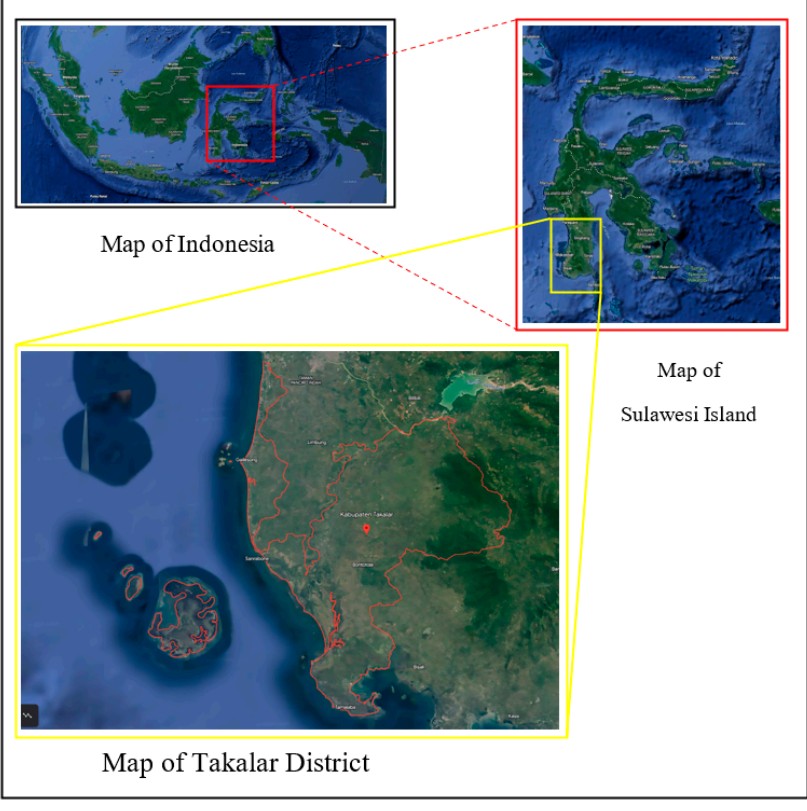

**Figure 2.** Map of the research site, Takalar District [29].

## 3. Results and Discussion

According to Article 1 point 11 [14] of the Indonesian Constitution No. 45/2009, individuals who engage in fishing activities as their primary means of subsistence, utilizing a boat with a maximum gross tonnage of five, are classified as small-scale fishers. This study is focused on small-scale fishers who possessed a boat during the data collection period, regardless of whether they owned the boat, borrowed or rented it at the time (Table 1). Three distinct categories of boats are commonly employed by fishermen: 1. a rowboat, 2. an outboard motorboat, 3. a motorboat (see Table 1). It is important to note that all three boat types fall under the 5 G.T. group. In order to discern and classify the individuals engaged in fishing activities according to their ownership of fishing vessels, they were categorized into six distinct groups based on the specific type of boat they currently possessed: 1. Fishers who owned motorboats and onboard motorboats (M.B. and O.M.B.), 2. Fishers who owned motorboats and row boats (M.B. and R.B.), 3. Fishers who owned a motorboat only (M.B.), 4. Fishers who owned outboard motorboats and rowboats (O.M.B. and R.B.), 5. Fishers who owned outboard motorboats only (O.M.B.), and the last six were fishers who owned rowboats only (R.B.) (Table 2). This classification was derived from the empirical evidence obtained in the respective research domain. The rationale for categorizing the fishers into six distinct groups was to elucidate their actual distribution of fishing boat ownership. The data revealed that 37% of individuals who engaged in fishing owned two distinct types of boats, whereas a small proportion of fishermen possessed multiple boats of each type, as seen in Table 2.

**Table 1.** Fishing boat's haracteristics.

| Type | Characteristic | | |
| --- | --- | --- | --- |
| | **Fishing Ground** | **Driving Force** | **Size and Price** |
| **Rowboat (RB) ***  | Shore (0–5 km) Single day fishing | Manpower (paddle) | Length is 2–4 m, width is 50–75 cm RP 700,000–RP 2,000,000 or USD 51–USD 145 ** |
| **Outboard Motorboat (OMB) ***  | Off-shore (0–20 km) Single day fishing | Single motor (can be removed or installed outside of boat before catching fish) | Length is 3–7 m, width 75–100 cm RP 8,000,000– RP 15,000,000 or USD 581–USD 1089 ** |
| **Motorboat (MB) ***  | Off-Shore (More than 20 km), Multi-day fishing | Single/double motor (installed permanently inside the boat) | Length is 10–15 m width 100–200cm RP 50,000,000– RP 100,000,000 or USD 3630–USD 7261 ** |

Source: sample survey conducted in Takalar District by the authors in February and March 2018. Note: * MB: motorboat, OMB: outboard motorboat, RB: rowboat, ** USD 1 = RP 13,771 (at the time of data collection).

**Table 2.** Fishers' boat possession.

| Category | Type of Boat and Ownership | | | |
| :---: | :---: | :---: | :---: | :---: |
| | | | Ownership | |
| | Type of Boat * | No. Fisher (No. Boat) | Type ** | No. Fisher |
| 1 | MB | 3 (1) | O<br>R/B | 2<br>1 |
| | | 2 (3) | O | 2 |
| | OMB | 4 (1) | O<br>R/B | 3<br>1 |
| | | 1 (3) | O | 1 |
| 2 | MB | 5 (1)<br>1 (2) | O | 6 |
| | RB | 5 (1)<br>1 (2) | O | 6 |
| 3 | MB | 38 (1) | O<br>R/B | 16<br>22 |
| 4 | OMB | 7 (1) | O | 8 |
| | RB | 7 (1) | O | 8 |
| 5 | OMB | 84 (1)<br>3 (2)<br>1 (4) | O<br><br>R/B | 71<br><br>17 |
| 6 | RB | 7 (1)<br>1 (2) | O | 8 |
| | Total | | | 152 |

Source: Sample survey conducted by the authors in February and March, 2018. Note: * MB: motorboat, OMB: outboard motorboat, RB: rowboat, ** O: Owned, R/B: Rented or Borrowed.

### 3.1. Characteristics of Fishers' Household

All the fishers who were interviewed were male. Approximately 93% of the individuals in question were in a state of matrimony. The mean age of fishers was 44 years, ranging from 22 to 82 years. There was no substantial difference in the average age observed among the users of different types of boats. The findings indicate that around 51% of those engaged in fishing fell within the age range of 15 to 44 years. Additionally, 32% of fishers were between the ages of 45 and 54, while the remaining 17% were aged 55 years or above. According to the World Health Organization [30], it has been recommended that the younger generation's involvement is crucial in promoting sustainable development. This principle also holds for preserving fishing livelihoods, as it is imperative to augment the proportion of young individuals engaged in fishing activities. Young fishers possess enhanced physical strength, exhibit adaptability toward novel innovations, and are motivated to enhance fishing livelihoods. The mean fishing experience of fishers was 22 years. The individual's engagement in fishing as an independent angler commenced when they began catching fish unaccompanied by their parents. It is plausible that throughout the period in which individuals assisted their parents, who were engaged in fishing activities, they did not perceive themselves as active participants in the fishing profession.

The average educational attainment among fishers is 5.5 years, with a significant proportion having completed merely elementary school. The education level in South Sulawesi Province in 2017, which is 10.5 years, is significantly higher than the observed value [31]. The financial strain experienced by households compels their children to partake in fishing activities at an early age to augment their earnings, resulting in a disregard for their educational pursuits. Based on the accounts of fishers, they began accompanying their fathers in the pursuit of fish at approximately ten years of age. As a result, their school attendance became irregular or, in certain instances, completely absent. A strong negative association existed between age and education level (Pearson test, r= −0.410, $p$ = 0.000

*p* < 0.01). It has been observed that younger individuals engaged in fishing activities exhibit a greater propensity for pursuing higher levels of schooling than their older counterparts.

The mean number of individuals per family is four, spanning from two to seven individuals. The mean number of individuals in the fishers' households was 2.04, ranging from one to five individuals. Within the fishery community, children are perceived as valuable resources that contribute to the sustenance of fishing practices, with particular emphasis placed on the role of sons. No statistically significant difference was observed in the means of household member, age, school term, and fishing experience across each category (one-way ANOVA test) (Table 3). The economic characteristics of fishers are presented in Table 4. The fishers with a motorboat exhibited the highest income, while those who relied solely on a rowboat had the lowest income. According to Table 4, there is a total of 61 individuals engaged in secondary employment as fishers.

**Table 3.** Social characteristic of fishers.

| Category | No. of Fishers | % | Household Member (People) | Age Years | Age Category 15–54 Years | Age Category >55 Years | School Period (Years) | Experience in Fishing (Years) |
|---|---|---|---|---|---|---|---|---|
| 1 | 5 | 3.3 | 4.2 | 45.4 | 3 | 2 | 7.2 | 23.8 |
| 2 | 6 | 3.9 | 4.5 | 42.0 | 5 | 1 | 5.0 | 23.7 |
| 3 | 38 | 25.0 | 4.1 | 41.2 | 34 | 4 | 6.4 | 16.5 |
| 4 | 7 | 5.3 | 3.6 | 44.3 | 5 | 2 | 5.7 | 19.9 |
| 5 | 88 | 57.3 | 4.5 | 45.1 | 73 | 15 | 5.1 | 24.8 |
| 6 | 8 | 5.3 | 4.5 | 45.1 | 6 | 2 | 3.75 | 18.0 |
| Average in total/Total | | | 4.2 | 43.9 | 126 | 26 | 5.5 | 22.1 |

Source: sample survey conducted by the authors in February and March, 2018.

**Table 4.** Economic characteristics of fishers.

| Economic Characteristics | Type of Boat RB (Cat. 6) (N = 8) | Type of Boat OMB (Cat. 4 and 5) (N = 95) | Type of Boat MB (Cat. 1, 2, and 3) (N = 49) | Average in Total (N = 152) |
|---|---|---|---|---|
| | (Thousand Rp./month) | | | |
| Primary job's income (fishing income) | 2100 | 4194 | 8017 | 5317 |
| Fishing variable cost | 1152 | 1214 | 2951 | 2366 |
| Fishing fix cost | 0 | 1436 | 2798 | 1795 |
| Fishing Net Income | 1072 | 1588 | 2351 | 2019 |
| Secondary job's income | 1627 (N = 3) | 1530 (N = 27) | 1207 (N = 31) | 1371 (N = 61) |
| Net income | 1682 | 2007 | 3105 | 3487 |
| Average food consumption expenditure | 1075 | 1233 | 1376 | 1271 |
| Average non-food household expenditure | 788 | 593 | 783 | 664 |
| Poverty line (USD/capita) | 1.21 * | 2.42 ** | 4.63 *** | |

Source: sample survey conducted by the authors in February and March, 2018. Note: USD 1 = RP and RP 13.771 = Indonesian Rupiah. * Extreme poverty, ** between extreme poverty and the poverty line, *** above poverty line (compared to World Bank Poverty line).

### 3.2. Fishers' Boat and Fishing Tools

As previously explained, fishers were classified into six distinct categories based on their possession of different types of boats. The fishers in this study possessed one or two types of boats, a characteristic closely associated with their financial capacities. The majority of individuals engaged in fishing activities could be classified into two categories: Category 3, which comprised 25% of fishers, and Category 5, which accounted for 57.3% of fishers. Notably, these categories exclusively consist of individuals who utilize either a

motorboat or an outboard motorboat for fishing purposes. According to Table 2, a total of 41 fishers, accounting for 27% of the sample, utilized rented or borrowed boats. As evident from the data presented in Table 5, fishermen who possessed a motorboat could reach more distant locations for fishing compared to fishers who relied solely on an outboard motorboat or a rowboat. There was a significant difference between mean number of fishing days ($p = 0.008$, $p < 0.05$), the sea fish catch amount $p = 0.005$, $p < 0.05$), distance to the fishing ground ($p = 0.004$, $p < 0.05$) and fish selling price ($p = 0.001$, $p < 0.05$) (One-way ANOVA test) (Table 5).

**Table 5.** Fishing activities-related information.

| Category | Fishing Day/ Month | Sea Fish Catch/ Month (kg) | Selling Place | Fishing Ground (km) | Selling Price/kg (Rupiah) | Average Revenue | Variable Cost | Fix Cost |
|---|---|---|---|---|---|---|---|---|
| | | | | | | (Thousand Rp./Month) | | |
| 1 | 24.8 | 258.0 | Local market and city market | 45.2 | 40,000 | | | |
| 2 | 22.7 | 195.3 | Shore and roadside | 23.6 | 45,416 | 8017 | 2951 | 2798 |
| 3 | 21.6 | 128.4 | Roadside and local market | 19.6 | 43,842 | | | |
| 4 | 21.7 | 178.6 | Home, shore and roadside | 18.4 | 30,071 | 4194 | 1214 | 1436 |
| 5 | 20.8 | 144.6 | Roadside and local market | 11.4 | 47,579 | | | |
| 6 | 25.5 | 174.5 | Home and shore | 7.3 | 14,875 | 2100 | 1027 | 0 |
| Average | 21.5 | 149.4 | | 20.9 | 43,782 | 5317 | 1759 | 1795 |

Source: sample survey conducted by the authors in February and March, 2018.

Fishermen engaged in fishing activities either individually or in the company of their family, relatives, or fellow fishermen, contingent upon the size of their fishing vessel, which determined their capacity to accommodate more individuals. The selection of specific fish species and fishing equipment, such as nets, that necessitate increased human labor is also a factor to be considered when attracting a larger workforce. Among the surveyed fishermen, it was observed that a range of fishing tools were possessed. Specifically, a small proportion of fishers (3.3%) did not possess any fishing tools, while the majority (62.5%) possessed only one type. Additionally, 13.8% of fishers possessed two types of fishing tools, 15.8% possessed three types, 3.9% possessed four types, and a mere 0.7% possessed five types of fishing tools. Fishers had a diverse array of fishing tools in their possession. Using diverse fishing instruments enables fishermen to effectively capture fish in various fishing grounds while also targeting certain species based on seasonal variations. Therefore, possessing a greater variety of fishing instruments increases the likelihood of successfully capturing more fish under different conditions.

*3.3. Fishers' Primary and Secondary Income*

Ninety-nine percent of the surveyed individuals who engaged in fishing reported that their primary occupation was fishing. The mean money generated from their principal occupation amounted to RP 5,316,493 (equivalent to USD 386.06) every month. The data show that 61 out of 152 fishers were engaged in secondary employment. Out of a sample size of 62 individuals engaged in fishing activities, 46.7%, 29%, and 18.7% of them reported having secondary occupations as farmers, seaweed farmers, and engaging in labor and self-employment, respectively. A mere 40% of the total population of 152 individuals who engaged in fishing activities also held a secondary occupation. Specifically, the individuals surveyed in this region engaged in the cultivation of seaweed and rice. In Tamala District, farming emerged as fishers' primary employment choice. The mean money generated from their secondary employment amounted to RP 1,370,819 (equivalent to USD 99.54) monthly. In addition to their primary fishing-catching occupation, fishers commonly participate in additional employment opportunities such as fisheries-related enterprises or seaweed cultivation. During periods characterized by strong winds, individuals engaged in fishing activities often described undertaking alternative forms of employment, such as manual labor or farming, due to the reduced frequency of fishing expeditions compared to the

regular season. Hence, it is customary for fishers to engage in a secondary occupation on a non-regular basis rather than monthly. Therefore, fishers rely heavily on fishing as their primary source of income.

To assess the poverty status of fishers in this study, the researchers employed the extreme poverty line (USD 1.90) and poverty line (USD 3.20) as specified by the World Bank in 2015. According to the calculation of the monthly net revenue derived from fishing activities and divided by the number of family members, it was shown that fishers utilizing rowboats had an average daily income of Rp. 16,706 (equivalent to USD 1.21) per person. This income level falls within the category of extreme poverty. Fishermen who employ outboard motorboats generated a higher daily income of RP 33,368 (equivalent to USD 2.42) per person, positioning them between extreme poverty and poverty thresholds. Fishermen who employed motorized boats exhibited higher incomes, earning a monthly income of RP 63,778 (equivalent to USD 4.63) per person daily, surpassing the poverty threshold. Therefore, it can be inferred that fishermen utilizing row boats and outboard motorboats in the Bacalar District experience poverty, while those employing motorboats have achieved economic sufficiency.

Based on Table 3, fishers belonging to Category 1 demonstrated the most extensive monthly capture, whereas those from Category 5 obtained the highest selling price. Fishers belonging to Category 6 exhibited the lowest selling price compared to other fishers, potentially due to their practice of selling their fish caught directly at home or along the shoreline. However, the fish catch of individuals in Category 4 surpassed that of fishermen in Category 5 due to their utilization of rowboats instead of the outboard motorboats used by the latter. However, due to the comparatively lower selling prices of their catch, the income generated by these fishers is likewise smaller than that of their counterparts. A lack of notable disparities is observed in the catch and selling price of marine fish across different categories. However, it is essential to note that significant distinctions exist in fishing income and the variable and fixed costs associated with fishing among these categories (one-way ANOVA test, $p = 0.000$ $p < 0.001$).

Prior to conducting regression analysis, several tests were performed, including assessments for linearity, heteroscedasticity, multicollinearity, and normality (assuming the Central Limit Theorem). The variables in the analysis were standardized. The correlation matrix for the research variables is presented in Table 6. Table 7 displays the outcomes of the inquiry conducted to examine the variables that impacted the net income of individuals engaged in fishing activities. This study utilized ordinary least squares (O.L.S.) regression analysis to examine the relationship between the net income of fishermen and other independent variables. The analysis incorporated thirteen distinct criteria, which are outlined in Table 7. This model demonstrated a substantial range in fishers' net income, as evidenced by an adjusted $R^2$ value of 0.624. There are five key aspects that exert a substantial influence on the net income of fishers. These factors include the type of boat utilized, the quantity of sea fish caught, the selling price of the fish, as well as fixed and variable costs.

**Table 6.** Correlation matrix of research variables.

| Variable | X1 | X2 | X3 | X4 | X5 | X6 | X7 | X8 | X9 | X10 | X11 | X12 | X13 | Y |
|---|---|---|---|---|---|---|---|---|---|---|---|---|---|---|
| X1 | 1.000 | | | | | | | | | | | | | |
| X2 | −0.429 ** | 1.000 | | | | | | | | | | | | |
| X3 | 0.640 ** | −0.384 ** | 1.000 | | | | | | | | | | | |
| X4 | 0.036 | −0.081 | 0.067 | 1.000 | | | | | | | | | | |
| X4 | −0.067 | −0.004 | 0.062 | 0.125 | 1.000 | | | | | | | | | |
| X5 | 0.132 | −0.176 * | 0.219 ** | 0.146 | 0.064 | 1.000 | | | | | | | | |
| X6 | −0.104 | −0.008 | 0.013 | 0.195 * | 0.362 ** | −0.013 | 1.000 | | | | | | | |
| X7 | 0.212 ** | 0.118 | 0.121 | −0.044 | −0.055 | −0.127 | −0.416 ** | 1.000 | | | | | | |
| X8 | 0.104 | −0.040 | 0.093 | −0.064 | 0.151 * | −0.152 * | 0.142 | 0.354 ** | 1.000 | | | | | |
| X9 | −0.033 | 0.155 * | −0.214 ** | −0.097 | −0.102 | −0.298 ** | 0.020 | 0.257 ** | 0.212 ** | 1.000 | | | | |
| X10 | −0.036 | 0.109 | −0.094 | 0.035 | −0.041 | −0.257 ** | −0.062 | 0.429 ** | 0.347 ** | 0.373 ** | 1.000 | | | |
| X11 | −0.011 | −0.095 | 0.088 | 0.072 | 0.186 * | −0.139 | 0.410 ** | 0.119 | 0.450 ** | 0.207 ** | 0.245 ** | 1.000 | | |
| X12 | −0.204 ** | 0.140 | −0.347 ** | 0.032 | 0.019 | −0.232 ** | 0.014 | −0.081 | −0.104 | 0.423 ** | 0.171 * | −0.049 | 1.000 | |
| Y | 0.054 | 0.072 | −0.005 | 0.102 | 0.222 ** | −0.210 ** | 0.555 ** | 0.282 ** | 0.270 ** | 0.248 ** | 0.118 | 0.181 * | 0.086 | 1.000 |

Source: sample survey conducted by the authors in February and March 2018. Note: * correlation is significant at the 0.05 level (2-tailed). ** Correlation is significant at the 0.01 level (2-tailed).

**Table 7.** Regression analysis of factors affecting fishers' net income.

| Variable | Description | Measurement | Mean | SD | Coefficient | t Value | Collinearity–Tolerance |
|---|---|---|---|---|---|---|---|
| Constant | Intercept term | | | | $1.275 \times 10^{-16}$ | 0.000 | |
| Age (X1) | Fishers' age | Years | 43.97 | 11.47 | 0.115 | 1.726 | 0.506 |
| School period (X2) | Formal school period attended by fishers | Years | 5.46 | 3.51 | 0.101 | 1.871 | 0.776 |
| Fishing experience (X3) | Fishers' fishing experience or how long they have been a fisher | Years | 22.07 | 11.92 | −0.011 | −0.162 | 0.470 |
| Household members (X4) | Number of household member living in the same house including the fishers themselves | People | 4.19 | 1.26 | 0.021 | 0.419 | 0.920 |
| Fishing days (X5) | Number of days spend for fishing in a month | Days/month | 21.47 | 6.11 | 0.017 | 0.327 | 0.835 |
| Boat category (X6) | Boat ownership category (Table 2) | Categorical data (1–6) | - | - | −0.109 | −2.133 * | 0.853 |
| Sea fish catch (X7) | Number of fish catch by fishers in a month | kg/month | 149.42 | 170.21 | 0.317 | 4.978 ** | 0.554 |
| Fish selling price (X8) | Selling price of fish per kilogram | Rupiah/kg | 43,782.89 | 41,379.83 | 0.175 | 3.285 ** | 0.786 |
| Fishing ground (X9) | Distance from the shore to the fishing ground | Km | 15.19 | 18.83 | −0.017 | −0.284 | 0.646 |
| Total net (X10) | Total number of net or catching tools possess by fishers | Nets | 1.58 | 0.97 | −0.013 | −0.229 | 0.671 |
| Fix cost (X11) | Fix cost spends by fishers for fishing activity in a month | Rupiah/month | 147,637.06 | 484,110.28 | −0.263 | −5.019 ** | 0.814 |
| Variable cost (X12) | Variable cost spends by fishers for fishing activity in a month | Rupiah/month | 1,736,538.27 | 3,576,308.00 | 0.610 | 8.934 ** | 0.481 |
| Second job (X13) | Availability of second job of fishers | Dummy (0 = No, 1 = Yes) | - | - | 0.032 | 0.551 | 0.663 |
| Net income (Y) | Total fishers' primary job and secondary job incomes minus variable cost and fix cost | Rupiah/month | 3,432,318.09 | 8,427,428.99 | | | |
| N | 152 | | | | | | |
| $R^2$ | 0.653 | | | | | | |
| Adj-$R^2$ | 0.624 | | | | | | |
| F-value | 22.411 | | | | | | |

Source: sample survey conducted by the authors in February and March, 2018. Note: * significant at 0.05 level, ** significant at 0.001 level.

The boat category had a negative significant impact on fishers' net income (Table 5); based on the one-way ANOVA test result ($p = 0.004$, $p < 0.05$) and the test of homogeneity of variances ($p = <0.001$, $p < 0.05$), it can be concluded that the mean of fishers' net income among the boat category groups was different. These results suggest that adverse effects could be attributed to the categorization of the boat. The fishers who owned motorboats and outboard motorboats were classified into Category 1 due to their possession of boats that were recognized for their superior quality and functionality. Fishers with both motor and row boats fell under Category 2. Due to two distinct considerations, fishers classified under Category 1 were deemed to enjoy a relative advantage in boat ownership compared to those classified under Category 2. Firstly, because of their ownership of O.M.B., individuals could travel the most significant possible distance for the sake of fishing. Additionally, in the context of short-distance fishing, the utilization of O.M.B. provided a distinct advantage in terms of fuel efficiency compared to the use of M.B. Based on this premise, fishermen classified in category 1 possessed a more excellent range of fishing locations, hence yielding a discernible beneficial effect on their ability to achieve higher catch volumes and reduce associated fishing expenses. The utility level of boats owned by fishermen diminished from Category 1 to Category 6. Category 6 comprised explicitly fishermen who possess

only a rowboat. Hence, it could be inferred from the correlation findings that an inverse relationship existed between the allocated category number and net income.

However, addressing the problem of fishing boats is a complex matter since it is intricately linked to the financial resources of fishermen in terms of their ability to acquire more advanced vessels. Several potential solutions could be implemented to enhance fishers' boat ownership. For example, one potential approach could be implementing a cooperative system to assist fishers financially. This assistance would be in the form of loans or credit services specifically intended to facilitate the purchase of a boat. Notably, this cooperative system could offer several advantages to fishers, including a low monthly payment rate, the absence of interest charges, and exemption from penalties in the event of late payments. Currently, fishers encounter difficulties when accessing loans or credit services from banks due to the requirement of providing collateral, such as land or home certificates, which is sometimes unattainable for many fishers. If fishermen have already sought a loan, it is likely that the loan amount is pretty small or that financial institutions might be hesitant to extend credit due to the elevated risk associated with such loans, given the potential challenges in repayment. Consequently, most fishermen are left with no choice but to seek financial assistance from either money lenders or community collectors (*papalele* in local terms).

Another potential solution is the revival or establishment of fishery associations. This is supported by the findings from interviews conducted with fishers, which revealed a lack of organized activities, training, or collective efforts in their fishing practices. The establishment of a fisheries association has the potential to offer assistance to fellow fishers. Within the organization, individuals can establish a savings system to obtain a boat for communal or individual utilization autonomously. This scenario could be implemented if all the individual members sell their whole fish catch to the fisheries association, selling the aggregated catch to the market at a higher price and with improved bargaining power. In this scenario, the profit is redirected to the fishers instead of being allocated to external entities. The effective management of aid and assistance from the fisheries department could be achieved by establishing a fisheries association. Rather than providing boats or nets to individual fishers, all members could rotationally utilize these resources. This approach allows for support a–o be expanded, providing benefits to a more significant number of fishers.

The capture of marine fish is a prominent determinant that exerts a favorable influence on the net revenue of fishermen. There are several factors contributing to the low fish harvest among fishers. These include unfavorable environmental conditions leading to decreased fish stock, inadequate fishing instruments, excessive fishing activity in a particular area, a restricted workforce or effort due to individual fishing practices, and other related factors. An additional investigation is required to ascertain the precise cause for the particular instance observed within the research location. However, one strategy that might be implemented to enhance the likelihood of increasing fish harvest is through the collaborative effort of engaging in communal fishing activities with fellow members of a fisheries association. By employing this approach, there is a greater likelihood of achieving a better fish yield on each fishing expedition.

The selling price was found to have a statistically significant positive impact on the net income of fishers. In most instances, fishermen offer their fish at a reduced price to *papalele* due to contractual agreements between the two parties. First, to address the potential exploitation of small-scale fishers during fish sales, the government must establish regulations related to the minimum price of fish in the market. Furthermore, it is advisable to promote the practice of fishers selling their catch in fish markets established by the fisheries department. This approach might enable fishers to sell their fish to consumers directly. The fisheries association could also provide a platform for fishers to promote their catch at a premium price. The fixed cost exerts a statistically significant negative influence on the net income of fishers. Under the fixed cost category, there are five specific factors to consider. These include purchasing a fishing boat and net expenses, such as credit costs,

interests, and depreciation value. Additionally, there are costs related to renting the boat and net and the maintenance expenses incurred for both the boat and net. In order to acquire a fishing boat and net, fishermen require a substantial number of financial resources. Often, they resort to borrowing funds from local collectors or money lenders, with only a small fraction able to secure loans through formal banking institutions, as previously indicated. When individuals resort to borrowing from village collectors and money lenders, commonly known as *papalele*, they participate in a money-lending contract system. Under this arrangement, borrowers are obligated to repay their loans at a significantly high-interest rate. Additionally, they are required to sell their fish catch to the lender. It is worth noting that, in many cases, the selling price of fish is lower than the prevailing market rate. This tendency exacerbates the already dire circumstances faced by fishers. In order to address these challenges, implementing capital sharing among fishermen through fishing associations or cooperative arrangements within the fisheries sector might present a viable alternative.

The variable cost showed a statistically significant positive impact on the net income of fishers. This implies that the current net income of fishers has not yet reached its full potential, as it has not yet encountered the law of diminishing returns. By enhancing variable costs, it is possible to optimize the net income of fishers. Six distinct components were categorized as variable costs, encompassing gasoline, ice, labor, food, bait, and general upkeep. Out of all the variables examined, it was found that only general maintenance did not exhibit a statistically significant link with variable cost, as determined by a Pearson correlation test. An additional investigation is warranted to ascertain the efficacy with which these goods are utilized in their fishing endeavors and to determine strategies that can enhance the cost-effectiveness of all these items, optimizing the profitability derived from fishing operations.

Finally, it should be noted that while the second job did not have a substantial impact on the net income of fishermen, the data collected throughout the study suggested that there were qualitative distinctions between fishers who relied solely on fishing as their source of income and those who engaged in secondary employment. Particularly noteworthy is the author's visit to a village where the inhabitants engaged in both fishing activities and seaweed cultivation. The housing conditions of the fishermen in that village were comparatively superior. Therefore, the implementation of supplementary sources of income for fishers presents a viable way to augment their earnings, particularly during inclement weather conditions that impede fishing activities. South Sulawesi was the most significant contributor (3,339,048 tons equivalent to 30.2% of the total seaweed production) of seaweed products (*Eucheuma cottoni* and *Gracilaria verrucosa*) in Indonesia in 2016 [32]. In the year 2016, the production of seaweed in Takalar District amounted to 1,034,305 tons. This production trend has been seen to consistently increase over the years. Notably, the seaweed production in the Takalar District accounts for almost one-third of the entire seaweed production in South Sulawesi Province [33]. Seaweed aquaculture exhibits significant potential for becoming a supplementary or primary source of income for fishers residing in the Takalar District, thus enhancing their overall welfare and alleviating them from impoverished circumstances. However, it should be noted that not all sea locations in Takalar District are currently suitable for seaweed cultivation due to various factors such as technological limitations, seaweed characteristics, and environmental conditions. Nevertheless, through the establishment of fisheries cooperation and fisheries association systems, the government and fisheries department could provide support in developing appropriate technologies to enhance the resilience of seaweed and expand cultivation to other sea areas. Additionally, they could offer assistance in providing essential resources such as seaweed lines, seaweed seeds, and extension services and training programs for seaweed cultivation. Therefore, individuals engaged in fishing activities could initiate their involvement in cultivating seaweed through aquaculture methods. In the present state of seaweed aquaculture in Takalar District, it is noteworthy that the regulation of seaweed aquaculture fields is currently without the government or fishery department authorities'

oversight. If effective regulation is implemented, the utilization of the seaweed aquaculture area could extend beyond the local fishers residing in the vicinity. This would enable fishers from neighboring regions to also engage in seaweed cultivation. Even in limited areas, such an expansion of cultivation opportunities could enhance fishers' income and ultimately contribute to their overall welfare.

## 4. Conclusions

In summary, the findings indicate that small-scale fishers, particularly those relying solely on row and outboard motorboats, experience poverty. The findings indicate that a fishing boat substantially influences individuals' income, with a positive correlation observed between boat advancement and income level. Three potential solutions could be implemented to enhance and broaden the livelihood of small-scale fishers. Firstly, the main approach involves the active involvement of government authorities (MMAFI, MAFD of South Sulawesi and Takalar District), N.P.O., and N.G.O. to facilitate the implementation of a small-scale cooperative finance or loan system for fishermen, as well as to administer capital distribution among them effectively. This initiative aims to enhance their fishing operations, such as improving fishing vessels and other related activities. Secondly, the revitalization or reestablishment of fishery associations with effective management systems can lead not only to fishers but also external stakeholders becoming involved, offering valuable assistance to small-scale fishermen in fostering collaborative efforts. The revival of fisheries associations is perceived as a crucial alternative for enhancing fishermen's livelihoods and living conditions, with the potential to foster the development of rural fishing communities. Thirdly, as anticipated in an occupation heavily reliant on natural resources, there is a significant decline in fish catch, particularly for small-scale fishers. Consequently, fishers must diversify their livelihood options by engaging in alternative income-generating activities, such as lobster, shell, fish, or seaweed aquaculture. This approach can facilitate the fishers' transition to other fisheries-related occupations. It would be advantageous if the management of this enterprise were undertaken by a fisheries organization operating as a community company.

## 5. Limitations

There exist certain limitations to this research study. The research conducted in this study utilized a limited number of samples due to constraints in time and funding during the data collection phase. This limitation is relevant to the application of the O.L.S. regression analysis. O.L.S. regression analysis is subject to various factors that can affect its results. These factors include influential observations, leverage points, and sensitivity to outliers. Additionally, the analysis is constrained by certain assumptions, such as homoscedasticity, normality, linearity, independence, and the absence of multicollinearity. Therefore, gathering a substantial quantity of data is imperative to obtain dependable outcomes. The author has tried to address all assumptions to minimize the potential for misinterpreting the results. Including additional data and samples could enhance the analytical rigor and overall quality of research on small-scale fishers' livelihoods. Furthermore, the comprehensive examination of fishers' supplementary employment details, employment status, and household members' aggregate income was constrained by the restricted duration of interviews conducted with fishermen. Additionally, it should be noted that these interviews were conducted on the beach rather than within fishers' residences in certain instances. Therefore, an additional investigation is required to augment the comprehensive understanding of the entire revenue of fishermen's households and how they are allocated among family members to meet their daily needs. Examining the household income of fishers could provide a more thorough perspective on their poverty conditions.

**Author Contributions:** Conceptualization, A.I.M.; Data Collection, A.I.M.; Data Process, A.I.M.; Writing—Original Draft Preparation, A.I.M.; Writing—Review and Editing, M.F., T.K. and M.S.; Supervision, M.F., T.K. and M.S. All authors have read and agreed to the published version of the manuscript.

**Funding:** The research was funded by the corresponding author privately and the university funded the publication costs.

**Informed Consent Statement:** This research has received a permit to conduct the research by the National Unity and Politics Takalar District, and was approved by District Office, Sub-District Office and Village Office in Takalar District where this research was conducted. Before conducting the interview, the author explained the purpose of the research and how the research would be caried out and consent from each participant was also obtained.

**Data Availability Statement:** Data obtained during the data collection process can not be shared publicly due to privacy and the agreement in the consent with respondents.

**Acknowledgments:** The author would like to convey sincere gratitude to all people who contributed to this research, including fishers and fisheries department officers who helped during the data collection and supervisors and colleagues who contributed during script preparation. This paper is submitted as a part of a doctoral study's requirement at the United Graduate School of Agriculture, Kagoshima University.

**Conflicts of Interest:** The authors declare no conflict of interest.

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
