# Peer review of "Small-Scale Marine Fishers’ Possession of Fishing Vessels and Their Impact on Net Income Levels: A Case Study in Takalar District, South Sulawesi Province, Indonesia"

_fishes, doi:10.3390/fishes8090463_

Round 1
Reviewer 1 Report
1. The sampling method and process are unknown: Line62-67 mentions that: In Takalar District there are 2,085 marine capture fishers in 2015. In Takalar district there are 527 row boat, 2,402 outboard motor boat, and 907 motor boat in 2016 and the production from capture fisheries amounted to 9,372 tons. In the case that the parent population was already known, although the convenience sampling method was used in the study, how to allocate the sample and why 152 were selected were not clearly discussed.
2. The data used is not clear: the research adopts cross-sectional survey design, but Line79-80 mentions that secondary data were collected from publication of previous research or secondary data sources. How secondary data is linked.
3. The definition of small-scale fishers is unclear: Line 85 mentions that this research is targeting small-scale fishers who has their own boat. Why these three types of fishing boats are defined as small-scale fishermen, and Line 87. They are divided into six categories of fishers based. Why are they divided into these six categories, whether the samples all have only one fishing boat (Category 3, 5, 6), or only one fishing boat each (Category 1, 2, 4), whether there are fishermen with three in the case of more than fishing boats, there may be more than three types of boats.
4. Line 103-104 mentioned "To support the 103 sustainability of fishing livelihood, the percentage of young fishers should be larger." Not sure why there is such a statement.
5. Table 3 lists Fishing activities, catch, selling price, income and cost, but Category 1 has the highest catch, and Category 3 or 5, which have higher fishing boat efficiency, does not have a higher catch. What is the reason?
6. In the regression analysis of Table 4, the dependent variable is the fisherman’s net income, and there is no relevant descriptive statistics for the data on the net income, and the questionnaire survey process is not clearly explained (statistical period), including fishermen’s perception of cost, and whether the memory of the catch or sale price is biased.
7. There is a significant negative correlation between net income and Category, variable cost, why? Line 203-204 mentioned that Boat category has significant impact on fishers' net income, the better their boat, the 204 higher their income. The results of the study are not clearly stated, nor are they consistent with the previous background information.
Moderate editing of English language required
Author Response
Dear Sir/Madam,
Thank you very much for your feedback. We really appreciate your valuable suggestions and corrections. We have revised the manuscript based on the feedback. We hope our revision can improve the quality of the manuscript.
We really hope our research can give a clear explanation of small-scale fishers' real conditions and can be a consideration for decision-makers in the fisheries sector in order to improve small-scale fishers ' livelihood conditions as well as their well-being.
Thank you very much for your help and consideration.
Sincerely yours,
Ahmad Imam Muslim (Corresponding Author)
Doctoral Course student,
The United Graduate School of Agricultural Sciences,
Kagoshima University,
Japan.

Reviewer 2 Report
The paper aimed to describe the real situation of small-scale fishers, analyzing fishers’ ownership of the boat, analyzing their income level to reveal their poverty condition, to analyze the factors affecting their income. The scope of the paper is really interesting. However, quality of writing is not satisfactory according to the journal requirements. Author can consider these comments to improve the paper.
General concept comments
1. Add some more recent literature in the Introduction section of the manuscript. Provide a systematic overview of the development of small-scale fisheries.
2. The potential mechanism can be further discussed. Clarify the mechanism of the impact of local fishers’ ownership of the boat on net income at the level of theoretical analysis.
3. The utilization of the statistical methods needs further clarification, and the interpretation of findings need to be cautious. OLS regression analysis has some limitations, such as multicollinearity among independent variables that can affect the explanation of the results.
4. Primary data was collected in 2018, and the authors can update the latest data. In addition, the sample size is small and needs to be further enriched to guarantee the significance and validity of the regression results.
5. As explained in the paper, fishers are categorized into six based on their boat type possession. At the end of the base-case regression analysis, heterogeneity analysis of different groups can be performed according to the previous categorization of fishers.
6. Proofreading throughout the article.
Specific comments
1. Formatting problems in the manuscript, references, tables, and figures should be edited. For instance, add the unit of the vertical axis in Figure 1. Integrate the contents of Figure 3 and Table 2 to form a complete descriptive statistical analysis. Simplify and correct the title of Table 3 to make it easier for the readers to understand.
2. The coefficient values in Table 4 are too large and most of the variables are not significant. The authors need to standardize the data before running the regression. Add the correlation coefficient calculations before the regression results to make it more complete.
Moderate editing of English language required.
Author Response

(The authors gave the same response as above.)

Round 2
Reviewer 1 Report
1.The number of samples in Category 4 and 5 in Table 2 and Table 3 is different. Is there an error? In addition, the number of samples in Category 3 or 5 is high. Taking Category 5 as an example, there are 88 fishermen with relatively high experience and high fishing frequency. Not significantly different from others, how can the reason be better explained?
2.The relationship between net income and Category shows a significant negative correlation. If explained by "while Category 1 is fishers who have Motor Boat and Outboard Motor Boat and Category 6 is fishers who only have Row boat, so fishers in Category 1 is better than fishers in Category 2-6 in boat type consideration.” is not very precise, and it is suggested to rethink how to express its meaning.
Minor editing of English language required
Author Response
Dear Sir/Madam,
Responses and 2nd revisions of the Manuscript
First of all, We would like to thank you for your feedback. We got some valuable and substantive suggestions or corrections from 1st and 2nd reviews of the manuscript to improve the article’s performance and quality.
We have tried to revise and address all the suggestions and corrections. Hopefully, the manuscript is of better quality and clearer now to the readers.
Please find the attachment.
Thank you very much. Have a nice day.
Ahmad Imam Muslim (Corresponding Author)
Doctoral course student, The United Graduate School of Agricultural Sciences,
Kagoshima University, Japan.

Reviewer 2 Report
The previous comments have been revised, but some minor revisions are suggested.
1. The analysis of potential mechanisms in the article is too concise.
2. The limitations of using OLS regression analysis could be explained at the end of the article.
3. There should be further editing of the detail in the graphs and tables. For example, there are two Tables 3 and two Tables 4 in your paper.
4. Proofreading throughout the article.
Minor editing of English language required.
Author Response

(The authors gave the same response as above.)
